# Vaccine discourse during the onset of the COVID-19 pandemic: Topical structure and source patterns informing efforts to combat vaccine hesitancy

Juwon Hwang[1]*, Min-Hsin Su[2], Xiaoya Jiang[2], Ruixue Lian[3], Arina Tveleneva[4], Dhavan Shah[2]

1 School of Media and Strategic Communications, Oklahoma State University, Stillwater, Oklahoma, United States of America, 2 School of Journalism and Mass Communication, University of Wisconsin-Madison, Madison, Wisconsin, United States of America, 3 Department of Electrical and Computer Engineering, University of Wisconsin-Madison, Madison, Wisconsin, United States of America, 4 Department of Marketing and International Business, Michael G. Foster School of Business, University of Washington, Seattle, Washington, United States of America

* juwon.hwang@okstate.edu

**Data Availability Statement:** The data relevant to this study are available on the OSF repository at

## Abstract

### Background

Understanding public discourse about a COVID-19 vaccine in the early phase of the COVID-19 pandemic may provide key insights concerning vaccine hesitancy. However, few studies have investigated the communicative patterns in which Twitter users participate discursively in vaccine discussions.

### Objectives

This study aims to investigate 1) the major topics that emerged from public conversation on Twitter concerning vaccines for COVID-19, 2) the topics that were emphasized in tweets with either positive or negative sentiment toward a COVID-19 vaccine, and 3) the type of online accounts in which tweets with either positive or negative sentiment were more likely to circulate.

### Methods

We randomly extracted a total of 349,979 COVID-19 vaccine-related tweets from the initial period of the pandemic. Out of 64,216 unique tweets, a total of 23,133 (36.03%) tweets were classified as positive and 14,051 (21.88%) as negative toward a COVID-19 vaccine. We conducted Structural Topic Modeling and Network Analysis to reveal the distinct topical structure and connection patterns that characterize positive and negative discourse toward a COVID-19 vaccine.

### Results

Our STM analysis revealed the most prominent topic emerged on Twitter of a COVID-19 vaccine was "other infectious diseases", followed by "vaccine safety concerns", and

https://osf.io/wbtc8/?view_only=
f534347d4b974747b3acc935b044d302.

**Funding:** This study was supported by grants from the University of Wisconsin - Madison Office of the Vice Chancellor for Research and Graduate Education with funding from the Wisconsin Alumni Research Foundation, the William and Flora Hewlett Foundation, and the John S. and James L. Knight Foundation. The funders had no role in study design, data collection and analysis, decision to publish, or preparation of the manuscript.

**Competing interests:** The authors have declared that no competing interests

"conspiracy theory." While the positive discourse demonstrated a broad range of topics such as "vaccine development", "vaccine effectiveness", and "safety test", negative discourse was more narrowly focused on topics such as "conspiracy theory" and "safety concerns." Beyond topical differences, positive discourse was more likely to interact with verified sources such as scientists/medical sources and the media/journalists, whereas negative discourse tended to interact with politicians and online influencers.

## Conclusions

Positive and negative discourse was not only structured around distinct topics but also circulated within different networks. Public health communicators need to address specific topics of public concern in varying information hubs based on audience segmentation, potentially increasing COVID-19 vaccine uptake.

## 1. Introduction

Development and uptake of a COVID-19 vaccine is a major step in fighting the spread of this novel disease [1], which has resulted in an unprecedented global public health burden. Herd immunity, which occurs when a sufficiently large proportion of a population has been vaccinated against or recovered from the specific infectious disease, is critical to slowing the pandemic's spread [2, 3]. To achieve herd immunity, it is estimated that a COVID-19 vaccine should be accepted by at least 75–80% of the population [4]. However, a recent study found a majority of the U.S. public would be uncomfortable being among the first to receive a COVID-19 vaccine and one-third of adults declining to accept a vaccine if offered [5], generating substantial concern from a public health standpoint [3]. To effectively address vaccine hesitancy and foster vaccine confidence [3], it is important to understand the nature of U.S. COVID-19 vaccine discourse.

Social media serve as a site for widely available and accessible public discourse surrounding the COVID-19 vaccine [6]. In contrast to traditional news media or scholarly outlets, social media content does not undergo an editorial processes or scientific vetting unless violating certain rules of a platform, thereby allowing users to voice their opinions on their own terms in most cases [6]. This characteristic facilitates users' ability to speak out on public health issues, such as vaccination, with no expertise required [7]. Indeed, vaccine discourse is frequently observed across social media platforms, with a considerable number of studies examining the different types of vaccine content on social media in contexts such as childhood vaccination schedules and HPV vaccines [8–14]. For example, content with negative sentiment toward vaccines was prominently present across social media platforms, with topics ranging from vaccine safety concerns [9, 10, 14], pharmaceutical and medical skepticism [9], conspiracy-style beliefs [9, 11, 12, 14], and infringement of civil liberties [13]. As such, a great deal of research has focused on online discourse that has *negative* sentiment toward vaccines [12, 14, 15], but little attention has been paid to *positive* counterparts, which provide a useful reference point to respond to concerns and topics voiced by users who hold negative opinions on vaccines. Few studies have considered positive vaccine content alongside negative vaccine content when analyzing social media; when they have, positive content was more likely to present accurate information [9] and emphasize vaccine effectiveness [16], compared to negative content. Moreover, due to the novelty of the COVID-19 pandemic, few studies have investigated the communicative patterns in which Twitter users participate discursively in vaccine discussions about this highly politicized issue.

Apart from the topical differences between the positive and negative vaccine discourse, tracking the specific groups in which the two discourses circulate not only clarifies the distinct characteristics between the entities, but also aids understanding of how positive and negative vaccine discourse is formed and shaped. Social media platforms, such as Twitter, allow users to interact with each other by following, sharing, and mentioning other accounts (i.e., @NAME). News feeds show self-selected streams for each account based on personal interests [6]. This process is further amplified by social media's proprietary algorithms. Consequently, the self-selection of networks as well as retweeting and @mentioning can be indicative of the flow by which positive and negative vaccine discourse spreads. While the effects of diverse media use on vaccination behaviors are well-documented [17–19], little is known about how contrasting vaccine discourses (i.e., positive vs. negative discourses) are spreading through different types of mentioned accounts.

This study has two aims. The first goal of the present study is to investigate the *content* of the Twitter discourse surrounding the COVID-19 vaccine during the four months after the declaration of the pandemic by the World Health Organization and widespread US lockdowns. We focus on this initial stage of vaccine development because public health communication during this time was inherently challenging due to high levels of scientific uncertainty, constantly changing information, and a highly partisan information environment [20]. Specifically, we compare topical prevalence of vaccine discourse that has either positive or negative sentiment toward a COVID-19 vaccine to understand distinct areas of concern with COVID-19 vaccines. Notably, we explore what individual users discuss regarding the issue, outside of the mere dissemination of elite-supplied messages. Second, we compare the types of *actors* with whom users who have either positive or negative sentiment toward the COVID-19 vaccine chose to interact.

By doing so, our study provides important insights and practical guides for public health communicators, which, to date, predominantly focus on disseminating vaccination guidelines or schedules, rather than proactively responding to vaccine hesitancy. Thus, we suggest public health communication professionals should broaden their messages to address specific concerns as revealed in public conversations on social media, as well as leverage a diversity of actors who have been centrally situated in the vaccine-hesitant communities.

## 2. Methods

### 2.1 Data collection

Given the purposes of this study, our analysis focuses on original tweets (i.e., tweets that were composed by ordinary users themselves). We constructed a unique dataset consisting of a random sample of 1% original tweets. To do this, we first scraped Twitter data through Synthesio, a social media monitoring tool to randomly sample 1% of public tweets using COVID-19 related keywords (S1 Appendix). Second, we retained only tweets that were topically related to vaccines using our customized dictionary (i.e., mentioned at least one of the keywords in the main text, excluding URLs), resulting in a sample of 349,979 tweets. Finally, to obtain tweets reflecting the user's interest and views, duplicated tweets were removed, including retweets (n = 251,025) and quote tweets (n = 34,738) following conventional practice [21], resulting in a final dataset of 64,216 user-generated COVID-19 vaccine-related tweets. Given that our analysis focuses on the different topics that positive and negative vaccine tweets emphasized when talking about a COVID-19 vaccine, we removed retweets that merely disseminate messages without user input; yet, quote tweets present an ambiguous case to determine the user's vaccine stance. Consider the following tweets that convey entirely different meaning but with similar languages: "This is a stupid idea! RT@Dr.Science: Vaccine is safe and effective" and "Such

an important point! RT @Dr.Science: Vaccine is safe and effective." The first tweet shows negative vaccine discourse, whereas the latter endorses vaccines despite the common language such as "safe" and "effective." Due to our focus on vaccine stance and topic detection, we opted for a more rigorous approach to avoid noise and reduce false positives. We acknowledge that user-initiated discussion constitutes only a subset of the entire twitter discourse, albeit an important one. We note that the patterns observed here largely hold when replicating the procedures with the complete dataset composed of both original tweets and retweets.

## 2.2 Stance classification

This study applied a supervised machine-learning technique to classify tweets into positive and negative discourse categories. Specifically, we used a transformer-based supervised machine-learning method, which allows labeling a tweet as expressing positive or negative sentiment regarding COVID-19 vaccines, and in turn, classifying the rest of the tweets based on the labeled samples using the Bidirectional Encoder Representations from Transformers (BERT) machine learning algorithm. There are several advantages of our current approach. Compared to other unsupervised textual analysis techniques, supervised machine-learning allows tracking discourses that are theoretically important or comparing discourses between groups on the same criteria. In our case, our supervised machine learning approach provides a computationally efficient way to classify tweets based on target-specific sentiments (i.e., sentiment toward vaccines), rather than more generic tasks of sentiment detection. By contrast, unsupervised machine learning is to learn the inherent structure of the data without pre-provided labels [22, 23]. Since the performance of the unsupervised learning-based sentiment analysis methods heavily relies on the pre-generated lexicon and corpus, if the sentiment/emotion relevant words do not exist in the lexicon or corpus, the classification accuracy will be low [22, 23]. Given that COVID-19 and its vaccines involve high levels of uncertainty and novelty, as well as its nature of constantly evolving, we believe that a supervised machine-learning approach would be appropriate to provide a scalable solution to text classification tasks with pre-defined criteria.

We defined positive vaccine discourse as tweets that express favorable sentiment or attitude toward a COVID-19 vaccine or contain affirmative information about COVID-19 vaccine development; negative vaccine discourse, on the other hand, was defined as those espousing unfavorable sentiment, commentary, or information about COVID-19 vaccines (see S2 Appendix for more details and examples).

To build a deep learning classifier, we first constructed a human-labeled set of tweets: two graduate students who were external from the research design, but had experience in general coding works and familiarity about the domain subject, were responsible for the labeling. After receiving training with the codebook developed by authors, each of them independently labeled a random sample of 200 positive and negative vaccine tweets (0 = absent, 1 = present). The classification problem with respect to each variable was treated independently, which allowed us to capture COVID-19 vaccine discourse types with more granularity. After achieving a sufficient level of intercoder agreement (Krippendorf's alpha = .90), the two coders each labeled another 5,000 randomly selected tweets and continued coding until a balance between the two classes was reached. The labeled tweets were then used to train and validate the stance classifier using the BERT machine learning algorithm, with dimension 768 embeddings and 5-fold cross-validation. The model performance has reached a satisfactory level of accuracy, 71% and 75% for positive and negative vaccine tweets respectively. After achieving satisfactory performance, the classifier was used to label the remaining tweets. Two coders manually verified the stance labels of a random sample of 200 tweets based on ML classification (93.5%

agreement). After removing tweets with neutral (24,497; 38.15%) or mixed (2,526; 3.93%) sentiment, we retain 23,133 (36.03%) tweets with positive sentiment and 14,051 (21.88%) tweets with negative sentiment toward the COVID-19 vaccine for structural topic modeling (More model assessment metrics are available in S3 Appendix).

## 2.3 Structural Topic Modeling (STM) analysis

To detect the topical structure of Twitter conversation surrounding the COVID-19 vaccine, we employed structural topic modeling (STM), an automated text analysis method that incorporates meta data into topic models [24]. Building on other traditional topic modeling techniques such as the latent Dirichlet allocation model (LDA), STM infers the latent topical structure based on word co-occurrence, using bag-of-word as the representation. Topics are "latent" in the sense that they emerge inductively as algorithms learn the hidden patterns underlying a collection of texts, offering the advantage of preventing researcher bias.

Unlike other probabilistic topic models that treat each document as a discrete observation, STM incorporates document-level information (i.e., whether a tweet has either positive or negative sentiment toward a vaccine), allowing for partial pooling of parameters along the structure defined by the covariates. We take advantage of STM's ability to incorporate meta-data in estimating topical prevalence in the positive and negative vaccine corpora. This allows us to not only find what Twitter users discussed, but also how their vaccine sentiment affected their tweeted topics.

Before conducting STM, several standard data-preprocessing procedures were taken to remove noises. These include the following: remove a) all non-English text and non-American Standard Code for Information Interchange (ASCII) characters, b) twitter handles, URLs, and common English stop words, and c) tokenization and lemmatization. As an additional data-preprocessing step, we retained words that are nouns, verbs, and adjectives, applying the part-of-speech (POS)-tagger from Stanford Core Natural Language Processing (NLP) suite. Focusing on specific parts of speech has been found to improve topic coherence and efficiently generate more coherent and meaningful (non-ambiguous) document clusters [25]. In creating the document-term matrix, we further removed too frequent (appearing in over 90% of the documents) or infrequent features (appearing in less than 0.005% of the documents), as their distribution patterns often do not contribute significantly to meaningful topics [26].

To obtain the optimal number of topics, we compared models with a broad range of possible k (2–100) on four commonly used metrics: coherency, exclusivity, residuals, and lower-bound. The resulting topics were labeled based on a) each topic's most frequently occurring features, b) top exclusive words that distinguished one topic from others (or FREX words), and c) the most representative texts (tweets with the highest theta scores) [24]. Two authors took steps to validate the topic labels with a random sample of 200 tweets (92% agreement).

## 2.4 Mention network

Beyond the distinct topical patterns, this study aims to further reveal how positive and negative vaccine discourse is circulating within certain types of networks. One of the ways to classify mentioned accounts through which vaccine discourses circulate is to focus on the degree to which account users are expected to provide medical expertise and/or how their posted content was associated with formal editorial processes [17, 18]. For example, a study classified health information sources as either official or unofficial sources based on the degree to which the source provides access to medical expertise and/or whether the content of the source undergoes vetted inspection [17, 18]. Based on this approach, we created primary categories (e.g., scientist/medical source, media/journalists, political source). We then identified the top

100 twitter accounts that have been mentioned by other users in our sample (see S4 Appendix) and manually assigned each a category label based on the literature. In this process, we recognized that there were a substantial number of accounts that didn't fall into the primary categories; thus, two new categories–online influencer and suspended account–were added. Therefore, we classified account mentions into five categories–scientist/medical source, media/journalists, political source, online influencer, and suspended accounts–with the first two categories based on the degree to which the source provides access to medical expertise, or whether the source is linked to an organization with formal editorial processes in place. Note that, as of February 2021, when this study was conducted, the former President Trump's Twitter account, "@realdonaldtrump" was suspended by Twitter due to the violation of their terms. However, since the account was active during our data collection (March to July 2020) and given his role as the President, we categorized this account as a political source rather than a suspended account.

Based on this classification, we examined the mention network of users' connections. In a mention network, a directed *edge* runs from A to B when A mentions B in a tweet. This communication practice creates a tie between two *nodes*, creating potential information pathways for tweeted content to flow between both users' networks (as a tweet mentioning another person will become visible in that person's timeline) [27]. Additionally, we conducted the Welsh t-test to statistically test whether positive and negative vaccine discourse resolves around different types of information sources.

## 3. Results

### 3.1 Content

**3.1.1 Volume of tweets.** We start by plotting the temporal dynamics of twitter volume during the course of our study time, with all types of tweets included, because the overall volume of tweets and how it evolved over time provides a big picture of our dataset before we delve into the examination of content of tweets. Overall, vaccine Twitter discussion saw a gradual uptick from March to May, with occasional fluctuations and a slight decrease after mid-May. As shown in Fig 1, public discussion about COVID-19 vaccines started to gain steam with the spread of the disease to society at large; the discussion peaked around mid-March, when the WHO declared COVID-19 a global pandemic and a potential timeline for vaccine development was released by the U.S. Department of Health and Human Services (HHS) [28]. The negative vaccine Twitter volume saw its first peak around mid-April, when President Trump criticized the WHO's handling of the pandemic and ordered a halt in its funding [29]. This was also around the time vaccine conspiracy theories surrounding Bill Gates started trending on social media, with topics like "microchip implants" and "ID2020."

COVID-19 vaccine discussions reached a second peak when Trump announced plans on March 15 for speedy vaccine development, manufacturing, and distribution in a timeline that maybe seemed overly ambitious [30], followed by the release of promising early results from Moderna's vaccine trials [31]. Overall, while vaccine discourse on Twitter saw several event-driven peaks, positive and negative vaccine discourse were prominent in different time period; positive vaccine voices remained more active in the early stage of the pandemic (i.e., March), negative vaccine ones were more salient at the lager stage of the study period (i.e., June to July). Our dataset of original tweets followed similar patterns.

**3.1.2 Prevalence of topics.** Next, we employed structural topic modeling to compare topical prevalence across positive and negative vaccine discourses. Table 1 showed the resulting 13-topic structure that characterized the U.S. COVID-19 vaccine discourse from March to

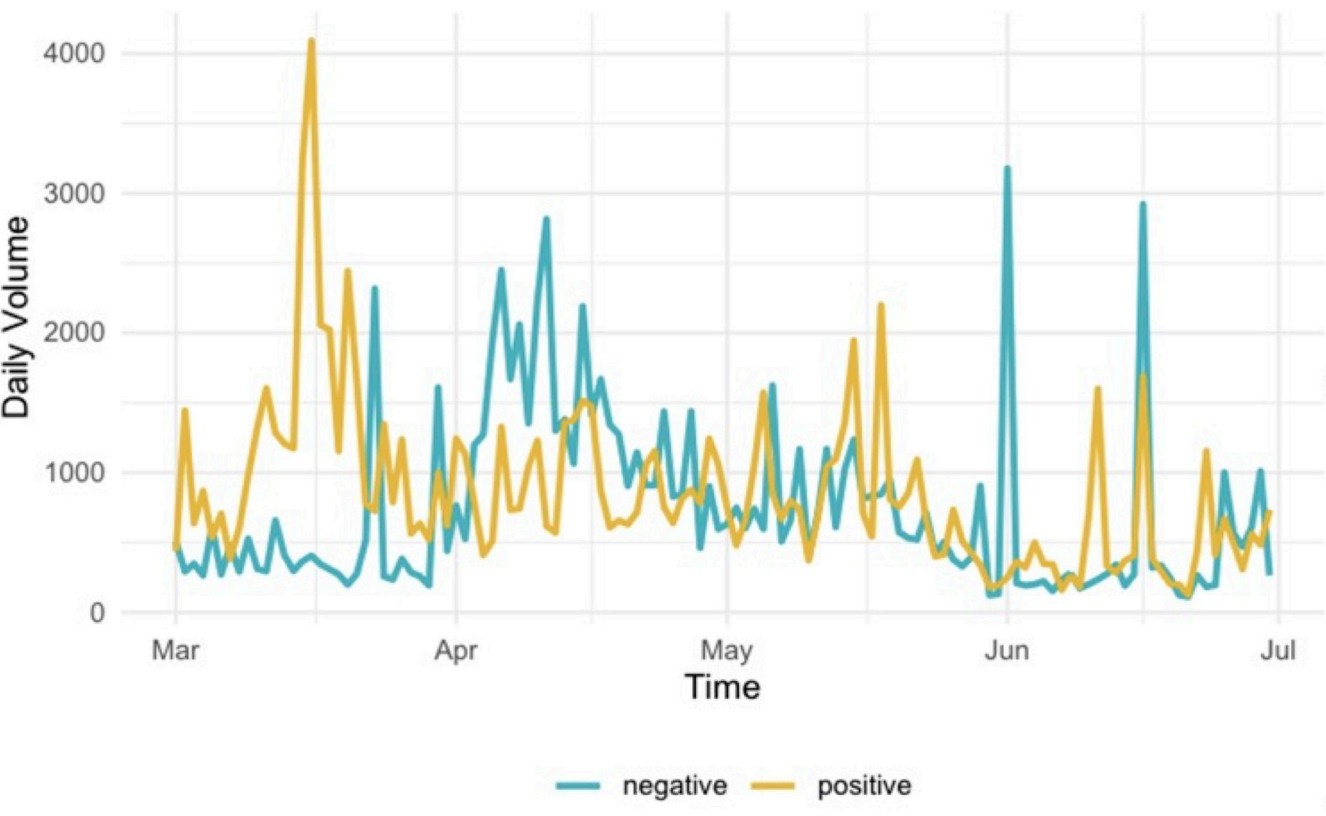

**Fig 1. Twitter discourse volume over time.**

**Table 1. The 13-topic structure that characterizes the U.S. COVID-19 vaccine discourses.**

| Topic | Label | Top Terms |
|---|---|---|
| 2 | Infectious Diseases | flu, death, die, spread, herd, immunity, outbreak, mortality, rate |
| 6 | Conspiracy Theory | push, force, mandatory, lie, chip, control, trust, implant, 5g |
| 3 | Safety Concern | child, kill, cause, kid, body, doctor, fact, inject, harm, injury |
| 9 | Inherent Uncertainty | flu, infection, disease, stop, risk, protect, immunity, prevent, symptom |
| 8 | Vaccine Development | trial, develop, research, scientist, dose, support, lead, mrna, #immunotherapy, #moderna, |
| 5 | New Normal | wait, public, economy, mask, family, school, allow, reopen, normal, business, closure |
| 7 | Consolidation and Mobilization | need, cure, help, antibody, hope, patient, medical, fight, ventilator, recover, supplies, focus, plasma |
| 1 | Monetary Motivation | money, drug, pay, fund, million, order, save, profit, bill, cost, patent, corporation |
| 11 | COVID Testing and Clinical Trial | testing, available, end, ready, phase, study, clinical trial, plan, approve, speed, accelerate |
| 10 | Vaccine Effectiveness | work, effective, safe, possible, science, home, future, social distancing, proven |
| 13 | Vaccine Safety Test & Production | test, company, create, candidate, safety, response, require, volunteer, production |
| 4 | Vaccine Information | health, change, fear, system, #vaccineswork, check, offer, law |
| 12 | Coping Strategies | treatment, potential, continue, global, product, result, provide, race, priority, basics |

June. Fig 2 shows the average gamma value for each topic (γ) (i.e., the estimated proportion of words from each document that are generated from that topic).

The most prominent vaccine-related topic in the initial phase of the COVID-19 pandemic in the U.S. was "other infectious diseases" (Topic 2), usually referring to the comparison of COVID-19 to other more familiar illness such as seasonal influenza in terms of fatality rate and transmission. Common narratives under this topic centered around arguments that the coronavirus "is not the flu" or "is just like the flu." The second and third prominent topics focused respectively on concerns over vaccine safety and potential side effects (Topic 3), often citing scientific studies or terminology, and conspiratorial explanations for mandatory vaccines or even the COVID-19 origin, playing up conspiracies surrounding "Bill Gates", "5G", "microchip implant", and "ID2020" (Topic 6).

Following these three topics were discussions over inherent uncertainty resulting from the novelty of the disease (e.g., implications for chronic health conditions) (Topic 9), news and updates about vaccine development (Topic 8), as well as disruptions of various aspects of social life (Topic 5). These daily disruptions range from school closure, stay-at-home orders, and restrictions on other business and social events (e.g., sports, bars, and restaurants). Tweets under this topic typically acknowledged the need for adjusting to the "new normal" before a safe and effective vaccine becomes widely available.

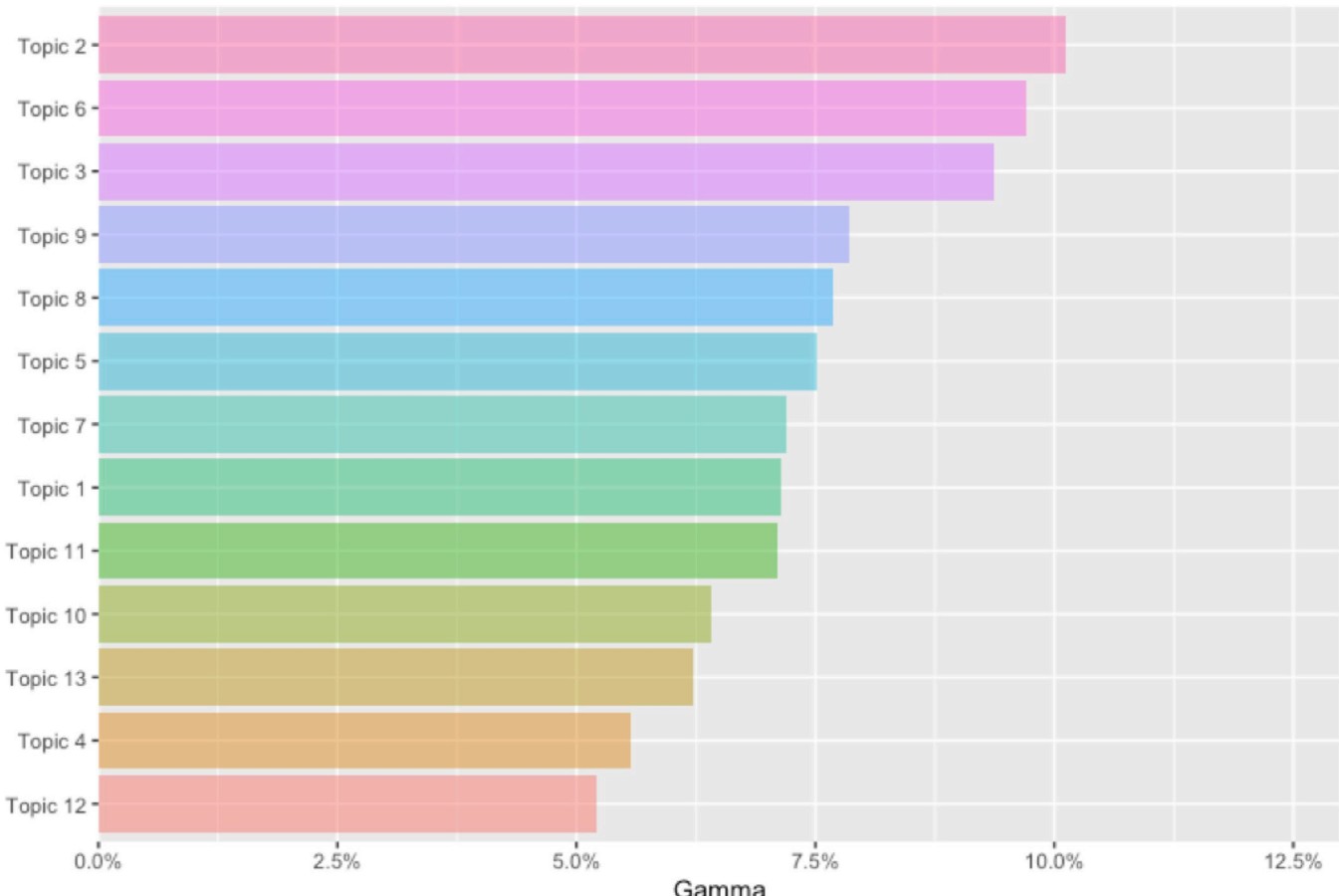

**Fig 2. The average gamma value for each topic (γ).** The document-topic probability, or the gamma value, is the estimated proportion of words from a given document that are generated from that topic.

Less prominent, but still constituting a sizeable proportion of Twitter discussion, were tweets that called for consolidation and mutual support (either in spirit or medical supplies such as PPE for medical professionals) (Topic 7), and those that cast doubt on the monetary motivations behind mandatory vaccination (Topic 1), often targeting institutions, mainstream media, and the political and financial establishment. A frequent reference point under this topic was the history behind the unpatented polio vaccine. There were topics more directly associated with vaccine itself, including safety testing and production (Topic 13), clinical trials (Topic 11), effectiveness (Topic 10). Finally, there was also a broader range coping strategies and reactions (Topic 12), and pop culture references or other types of "soft news" (Topic 4).

**3.1.3 Topic communities.** To further explore the relationships among topics, we constructed a semantic network in which topics serve as nodes and their associations as edges, using pairwise cosine similarity calculated based on the theta-cosine matrix generated by STM [32]. Topics were further grouped into clusters using Spinglass, a widely used community detection algorithm [33]. The association between two topics (i.e., nodes) is reflected by the thickness of the undirected line connecting them (i.e., edge). The edges are weighted, calculated based on cosine similarity, with the weight indicating topic co-occurrence in documents. Specifically, the higher the weight, the more likely that two topics are discussed within a given document. For example, vaccine production is more likely to be talked about alongside other coping strategies, yet it has a low cosine similarity value with the topic of conspiracy [32].

Our results suggest two distinct topic communities (see Fig 3), with one associated with the COVID-19 vaccine itself, ranging from clinical trials (Topic 11), development (Topic 8), manufacturing and distribution (Topic 13), to vaccine effectiveness (Topic 10) and calls for medical support and global cooperation (Topic 7). The second cluster revolved more around public deliberations on several contentious issues, scientifically based or not, as people made attempts to understand the novel disease by comparing it to other infectious diseases (e.g., Influenza, Polio, MMR) (Topic 2), reckoned with the disease's impact on social lives (Topic 5), and created simplified narratives for societal crises linked to the pandemic (Topic 1 and Topic 6).

**3.1.4 Relative volume of topics over time.** Finally, to uncover the overtime trend, we plotted the relative tweet volume over the four-month period. This approach helped us investigate the patterns in which each topic gained (or lost) prominence relative to others, controlling for the fluctuation in public attention to the COVID-19 vaccine overall. As shown in Fig 4, Topic 2 (Infectious Disease) dominated early Twitter vaccine conversation, alongside expressions of safety concerns (Topic 3). While some topics received more sustained overtime attention (e.g., Topic 6 –Conspiracy Theory), others showed a higher degree of fluctuation in Twitter volume (e.g., Topic 8 –Vaccine Development).

**3.1.5 Distinct topical prevalence in positive vs. negative valenced tweets.** To understand the discourses of Twitter users with positive and negative perspectives on COVID-19 vaccines, we compared topical prevalence across tweets with different viewpoints on vaccination (see Table 2 and Fig 5).

Overall, positive vaccine discussions demonstrated a wider range of perspectives, with greater attention to vaccine research updates, progress (e.g., clinal trial and safety test), and production. Additionally, compared to negative vaccine discourse, positive vaccine discussions devoted more Twitter conversation to general coping strategies and the importance of adjusting to the "new normal" before a vaccine becomes available, by taking preventive measures such as social distancing, washing hands, and wearing masks. There was also a greater emphasis on supporting medical professionals (e.g., provide PPE for health care workers while waiting for vaccines), calls for global collaboration (e.g., #inThisTogether, #defeatdiseasetogether), and comparisons between the coronavirus and other infectious diseases. Finally, while positive

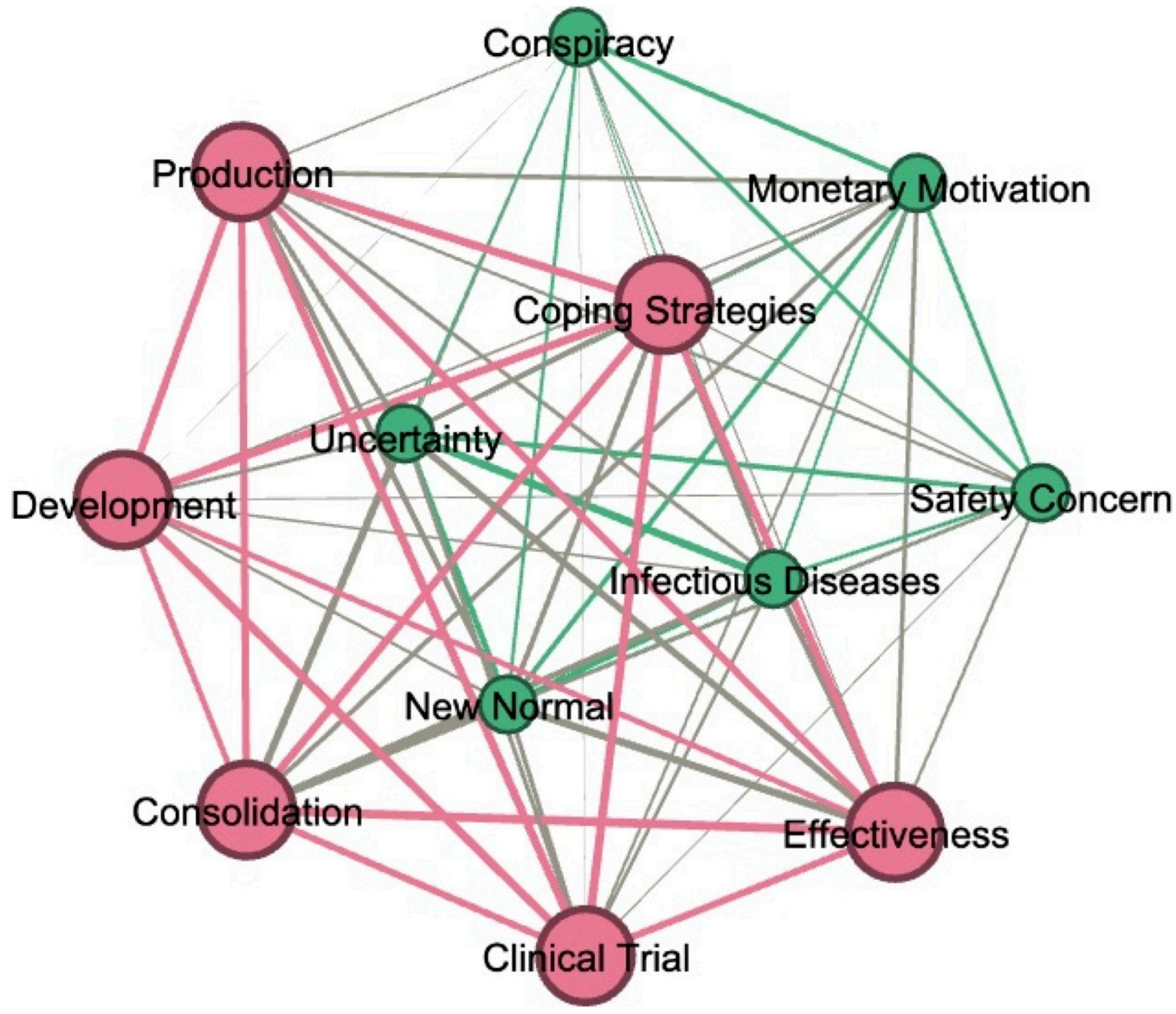

**Fig 3. A semantic network with nodes as topics and edges as their associations.**

vaccine tweets were more likely than negative vaccine ones discuss vaccine effectiveness, they also paid significantly more attention to the inherent complexities and uncertainty associated with the COVID-19 vaccine, such as implications for those with chronic health conditions.

Compared to the topical diversity in positive vaccine discourse, we see negative vaccine Twitter conversation dominated by a narrower set of narratives. Negative vaccine discourse was more likely to be about the monetary motivations behind mandatory vaccines, concerns over side effects, as well as conspiratorial beliefs.

### 3.2 Actors

**3.2.1 Mention network.** The mention network analysis also revealed distinct interaction patterns among positive and negative vaccine Twitter users (see Fig 6). To construct the

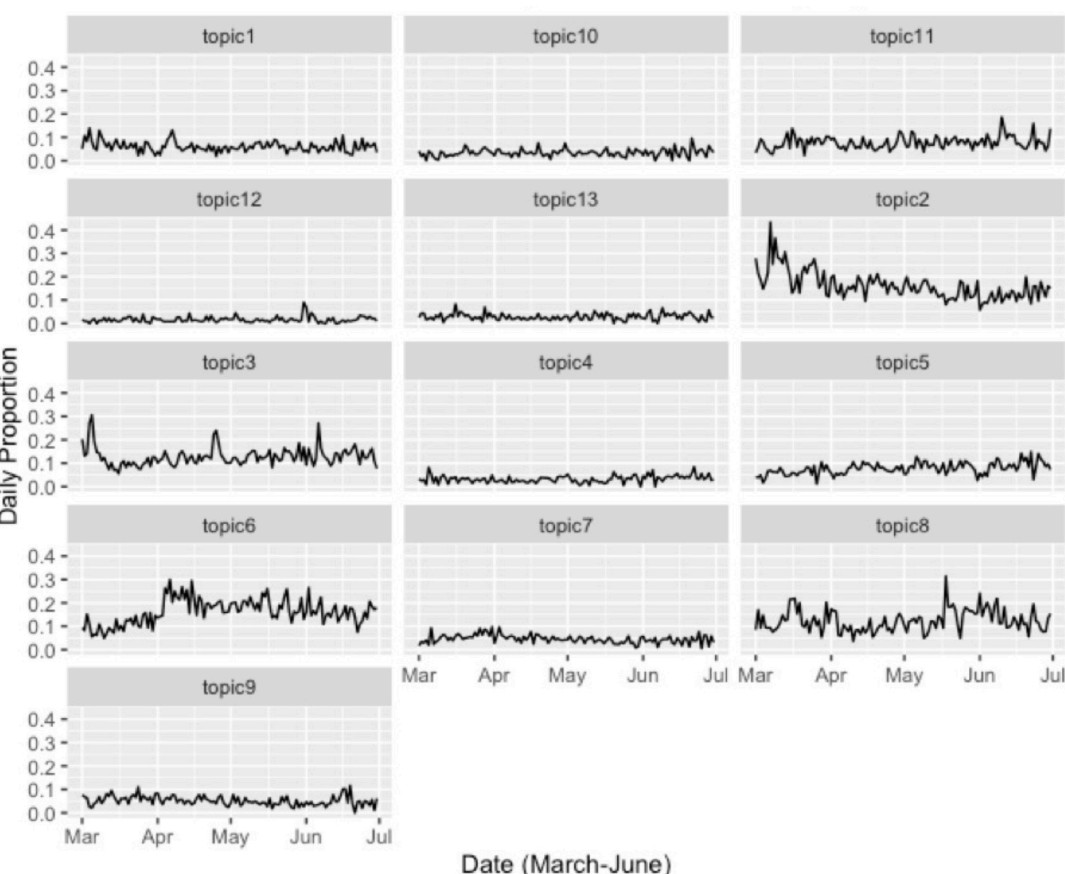

**Fig 4. Overtime trend in daily relative volume by topics.** Topic1 = monetary motivation, Topic 2 = infectious disease, Topic 3 = safety concern, Topic 4 = vaccine information, Topic 5 = new normal, Topic 6 = conspiracy, Topic 7 = consolidation, topic 8 = vaccine development, Topic 9 = inherent uncertainty, Topic 10 = vaccine effectiveness, Topic 11 = clinical trial, Topic 12 = coping strategies, Topic 13 = vaccine production.

**Table 2. Regression analysis predicting topical prevalence by positive and negative vaccine stance.**

| Negative vaccine sentiment (1)[a] | **Topic 1** | **Topic 2** | **Topic 3** | **Topic 4** |
|---|---|---|---|---|
| | Monetary Motivation | Infectious Diseases | Safety Concern | Vaccine Info |
| | .07[b] | −.03[b] | .12[b] | −.00 |
| Negative vaccine sentiment (1) [a] | **Topic 5** | **Topic 6** | **Topic 7** | **Topic 8** |
| | New Normal | Conspiracy | Consolidation | Development |
| | −.02[b] | 19[b] | −.04[b] | −.07[b] |
| Negative vaccine sentiment (1) [a] | **Topic 9** | **Topic 10** | **Topic 11** | **Topic 12** |
| | Uncertainty | Effectiveness | Clinical Trial | Coping Strategy |
| | −.01[b] | −.05[b] | −.07[b] | −.04[b] |
| Negative vaccine sentiment (1) [a] | **Topic 13** | | | |
| | Production | | | |
| | −.02[b] | | | |

[a]Reference: positive vaccine discourse (0)

[b]$P < .001$.

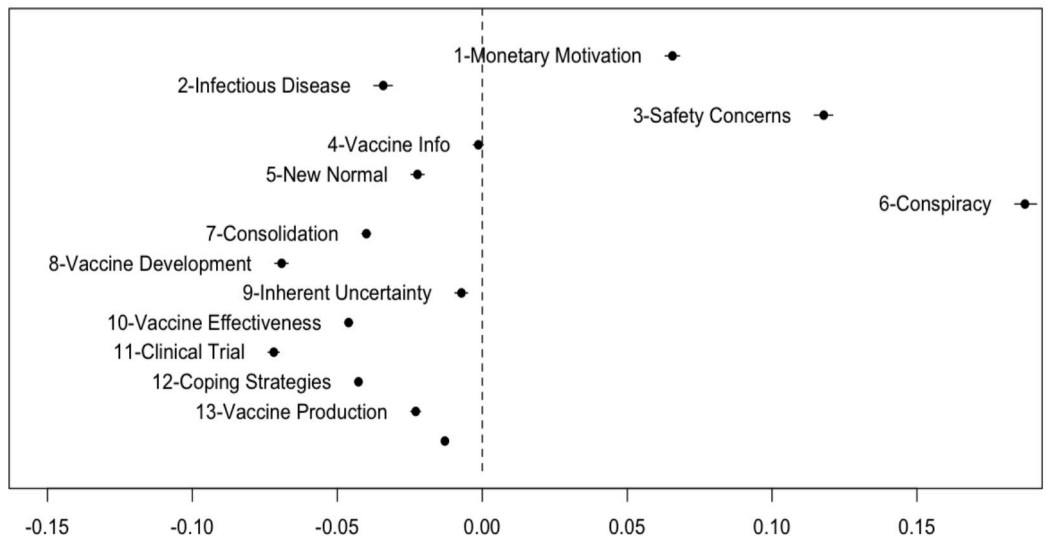

**Fig 5. Difference in topical prevalence.** Moving to the left means positive- and moving to the right means negative vaccine Twitter discourse.

mention network, we first examined the top 50 most mentioned accounts in positive and negative vaccine Twitter discourses, resulting in the 100 most influential users in total (see S4 Appendix). When an account was classified into more than two categories (e.g., a doctor who is an online influencer), the priority was placed on 1) suspended account, 2) scientist/medical source, 3) political source, 4) media/journalist, and 5) online influencer.

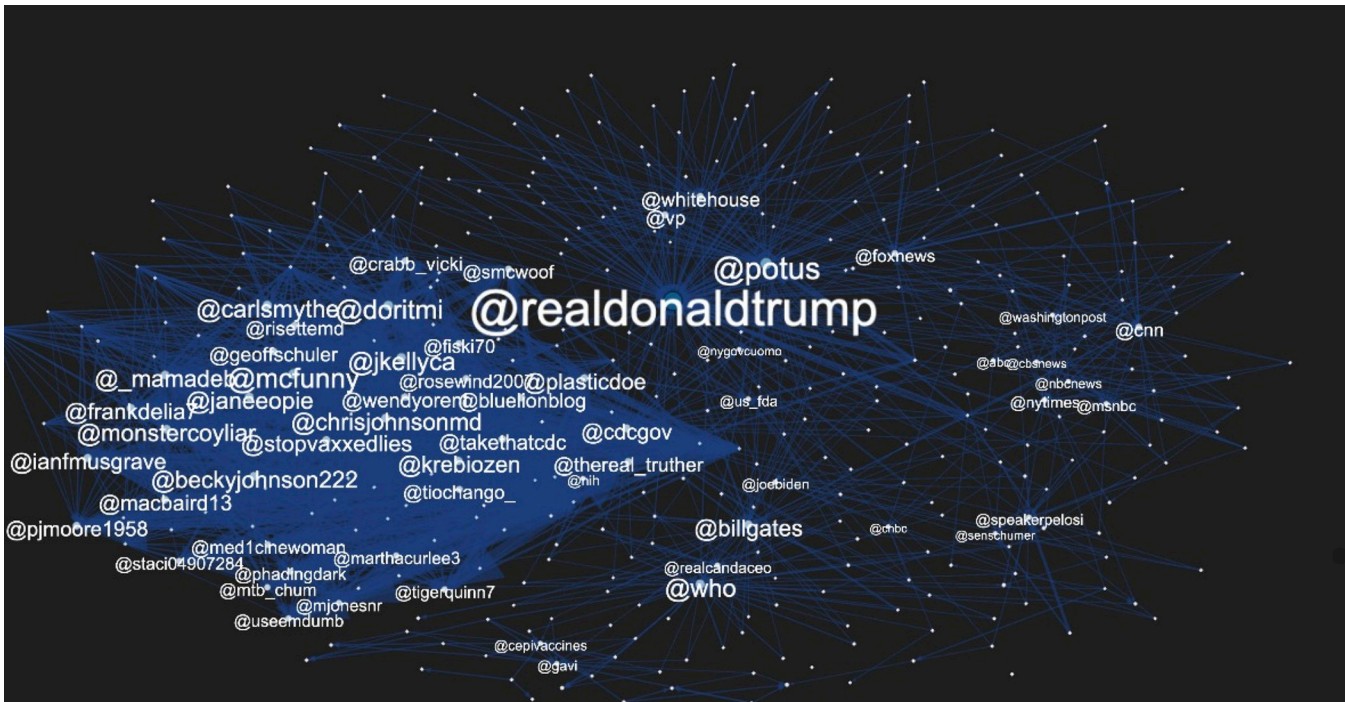

**Fig 6. The mention network.**

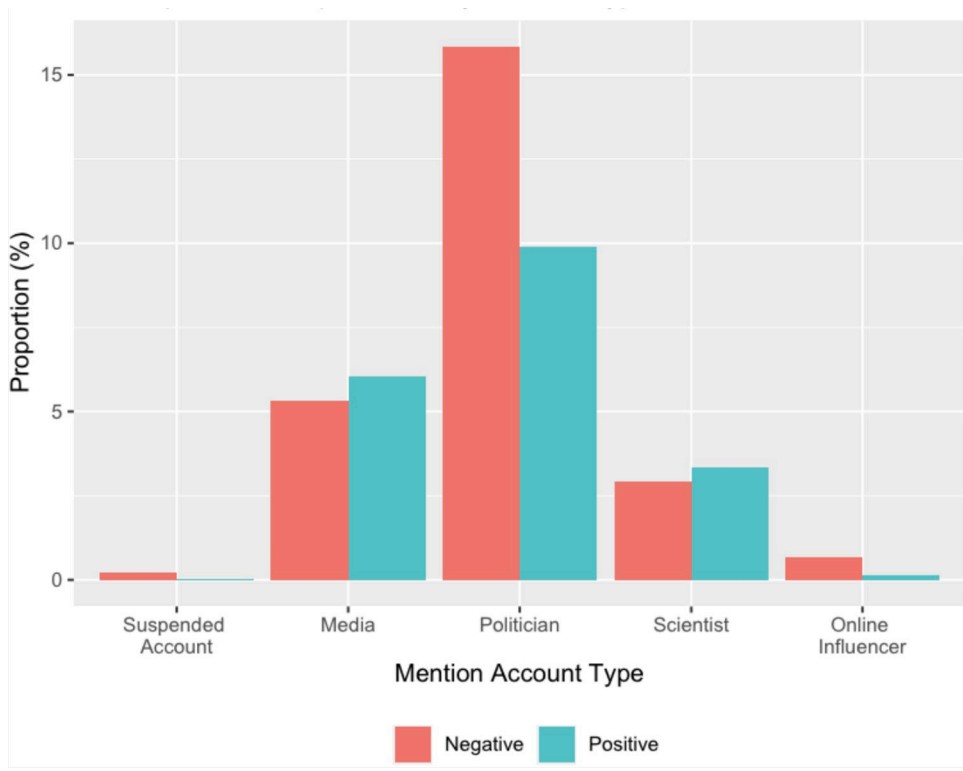

**Fig 7. Proportion of the top mention by account type and discourse category.**

Overall, positive and negative vaccine discourses tended to mention distinct types of Twitter accounts. Over half of the most mentioned accounts were non-overlapping, prominent only in one group (58%). Further, among the co-mentioned influential users, political actors were the most mentioned type (38%; @realdonaldtrump, @joebiden, @berniesanders, @whitehouse, @potus, @speakerpelosi, @janeeopie, @doritmi), followed by media accounts (23.8%; @cnn, @foxnews, @nytimes, @thehill, @realcandaceo), online influencers (19%; @mcfunny, @monstercoyliar, @frankdelia7, @beckyjohnson222), and public health institutions (9.5%; @who, @cdcgov). The top co-mentioned accounts also included Bill Gates (@billgates) and one suspended user (@jkellyca).

**3.2.2 Mention by account type.** There was a noticeable difference in how each type of account mention was able to get attention from positive and negative vaccine users (see Fig 7). Specifically, while media/journalists and scientists/medical sources tended to get mentioned in positive vaccine tweets, political figures, online influencers and suspended accounts appeared more influential among users who have negative sentiment toward the COVID-19 vaccine.

To confirm the above patterns, a series of Welch's Two-Sample $t$-tests were performed. Welch's $t$-test was chosen because it provides an alternative to traditional $t$-test for samples with unequal size and variance [34]. Also, to account for confounding factors such as tweet length or the difference in each group's tendency to use @ functionality, we calculate the average mention counts for each source type at the tweet level, which gives an indicator of the relative attention a tweet devoted to a particular source type (e.g., scientists/medical sources) relative to others. Results confirm that positive vaccine discourse was more likely to mention scientists/medical sources, $t(19892) = 2.54$, $P < .01$ and media/journalists, $t(20013) = 4.52$, $P < .001$; by contrast, negative vaccine tweets tended to interact with political sources, $t(18194) = -$

13.68, $P <$ .001, online influencers, $t(12425) = -10.21$, $P <$ .001, and suspended accounts, $t(12425) = -5.58$, $P <$ .001.

## 4. Discussion

Our study focused on positive and negative vaccine sentiments on tweets to understand the nature of public involvement as one of the key forces shaping vaccine acceptance and policy making. Specifically, we examined a) *content* difference in positive versus negative tweets surrounding the COVID-19 vaccine, as well as b) prominent *actors* frequently mentioned by the two camps.

The primary findings were summarized as follows: As to content difference, while positive vaccine discussions demonstrated a wider range of perspectives informed by research and medical professionals, negative vaccine conversation was dominated by a narrower set of narratives, such as monetary motivations behind mandatory vaccines, concerns over side effects, as well as conspiratorial beliefs. Furthermore, while positive vaccine discourse interacted with verified agents, negative vaccine discourse tended to mention online influencers and suspended accounts, suggesting possible pathways through which misinformation was spreading. Regarding the difference of prominent actors between the two camps, while positive vaccine discourse tended to be circulated by the network that consisted of scientists/medical sources and media/journalists, negative vaccine sentiment tended to be spread by another network including political sources, online influencers, and suspended accounts.

### 4.1 Content

Our STM findings on the prominent topics surrounding a COVID-19 vaccine revealed the way in which Twitter users attempted to understand this novel illness and the vaccine being developed to treat it. That is, users made comparisons between COVID-19 and other infectious diseases such as Polio, MMR, HPV, and most notably, influenza. For example, users discussed how the development and uptake of existing vaccines have successfully controlled the spread of infectious disease (e.g., "countless people in the world were infected with A disease and died until A vaccines were developed"). Importantly, when the same terms (e.g., influenza) were used, the context of use could be entirely different [35]. For example, users with negative attitude toward the COVID-19 vaccine, despite relatively low levels of activity in this topic, also talked about other infectious diseases, especially when they downplayed the severity of the COVID-19 (e.g., "you don't know that seasonal flu is deadlier than coronavirus"). Indeed, false comparisons with other diseases were among the most prevalent myths on Twitter in the early stage of the pandemic [20].

While positive vaccine discourse covered a wide range of topics from vaccine narratives (e.g., vaccine development, effectiveness, safety test) to the new normal of daily life (e.g., coping strategies for COVID-19, other preventive measures, appreciation for medical professionals), negative vaccine discourse was narrowly focused on topics such as conspiracy theories and safety concerns. The discourse suggested that users with positive vaccine stance may adopt diverse viewpoints in navigating the COVID-19 vaccine issue, whereas users with negative vaccine stance tend to propagate existing anti-vaccine narratives, often in the form of unmistakably false claims including conspiracy theories, unchecked rumors, false prevention methods, and dubious cures [20] while reinforcing the old topics in the new context of COVID-19 [36]. Indeed, the prevalence of conspiratorial thinking related to COVID-19 is unsurprising given that rumors and conspiracy theories about other outbreaks of infectious diseases have long prevailed [36]. The narratives of conspiracy surrounding COVID-19 featured secret plots that connected powerful individuals (e.g., Bill Gates) or institutions to inflict intentional harms or

prolonged surveillance. These conspiracy narratives were less disease focused but more politically motivated agendas (e.g., 5G Wireless, chemtrails, depopulations, etc.), all of which were designed to rouse fear and limits public's willingness to get a COVID-19 vaccine. These results imply that public health communicators need to design effective messages to prevent the public from building mental connections between existing "old" vaccine myths and the novel disease, especially among less receptive audiences.

Another predominant topic in negative vaccine discourse was concerns over vaccine safety and potential side effects. Some tweets in this topic emphasized the risk of the vaccine while downplaying the risk of the disease (e.g., "Learn the risk of vaccine! Even before the advent of the vaccine, the mortality of disease had no difference"). Whether a conscious or unconscious decision, vaccination does require weighing the risks of the disease that the vaccine is designed to prevent against the risks of vaccine side-effects [37]. Due to the rapid development of a COVID-19 vaccine and uncertainty of long-term side effects, discourses emphasizing the risks of a COVID-19 vaccine potentially outweighing the risks of the COVID-19 disease were circulated, including by legitimate sources. Yet many of these users were likely to argue that their personal perceived susceptibility to COVID-19 is quite low, but the general risk of the vaccine is high. Communication professionals need to assist the public in interpreting risks by providing useful points of references for legitimate risks vs false hysteria.

### 4.2 Actors

Mentioning other accounts in the posts may be an important function in spreading positive and negative vaccine discourse and potentially growing the two different networks. Our findings revealed there are two different networks that diffuse the distinctive views on a COVID-19 vaccine. The first network that tended to circulate positive vaccine discourse consisted of scientists/medical sources and media/journalists. This network is viewed as credible resources in public health due to their medical expertise or gatekeeping process [17, 18].

By contrast, another network that tended to spread negative vaccine sentiment included political sources, online influencers, and suspended accounts, defining this network as unverified resources. As online influencers are people who establish online profiles and voice opinions based on a topic with which they are familiar [37, 38], the content produced by online influencers is by nature subjective and often times extreme. In addition, political sources are increasingly polarized regarding vaccination, usually based on their ideology [39]. Public health issues are often politicized [39], with the COVID-19 vaccine a prime example of this growing phenomenon. Finally, given Twitter's effort to limit the spread of misleading and false health claims [40], suspended accounts may be seen as hubs of misinformation. All of these actors in this network show that negative vaccine discourse is circulated within a closely connected network of dubious sources lacking information gatekeeping.

### 4.3 Limitations and future directions

Our study has limitations in several aspects. First, we used a random sample of data generated by a particular data collection platform Synthesio. While this is an established way of collecting data, it is not certain whether and how the sample would have bias. Similarly, since we focused only on original tweets to detect prominent topics, caution should be exercised in generalizing these findings to the entire body of tweets. We, however, opted for a more rigorous approach to avoid noise and reduce false positives in vaccine stance and topic detection. Second, this study focused on the first four months of the pandemic, which limits the ability to make conclusions about temporal patterns of topical prevalence throughout the ongoing pandemic. We chose to focus on this initial phase of vaccine development because public health

communication was faced with unique challenges due to high levels of scientific uncertainty, and constantly changing information, and politicized environment [20]. Future research should extend the time period to draw a more complete picture of topical variance and temporal dynamics throughout the pandemic.

### 4.4 Practical implications for public health and vaccine communication

Our findings provide important insights for vaccine communication during the pandemic. Specifically, our results highlight the type of *content* that needs to be addressed to improve vaccine hesitancy as well as *actors* that need to be targeted as a conduit through which negative vaccine content diffuses, when disseminating vaccine information in order to increase the COVID-19 vaccine uptake rate. For effective *content* to promote the acceptance of a COVID-19 vaccine, public health and vaccine communicators could harness social media with positive stories of vaccination experiences that may be able to shift vaccine perceptions among vaccine hesitant individuals. Alternately, false, misleading, or otherwise negative content could be detected and countered on an individual or mass scale. Clearly, a substantial number of tweets in Topic 3 (concerns over vaccine safety) cited stories of people who experienced adverse reaction to vaccines as their reason for believing the risk or side effects of vaccines. Given that people are strongly swayed by personal narrative, and that those stories have a strong power to alter perceptions of risk [41], it is necessary to make positive vaccination narratives visible on social media platforms.

When it comes to the *actors* who need to be targeted, our pattern of results, in which negative vaccine discourse was predominantly circulating within an online influencer-centered network, suggests that effective intervention should also attempt to shift the dominant source of information and interactivity patterns among Twitter users who have negative views on the COVID-19 vaccine. Notably, these actors are, despite their lack of medical expertise or vetted inspection [17, 18], considered a reputable source within many networks. Thus, partnering with online influencers could be beneficial for public health communicators to reach their followers, though encouraging involvement from this group poses its own challenges. Communication efforts may involve prominent information hubs by tagging or mentioning influential accounts. By doing so, users who have negative sentiment toward vaccines are more likely to be exposed to science-based information, even just incidentally.

## 5. Conclusion

While a great deal of research has focused on vaccine hesitancy online, less has paid attention to both positive and negative vaccine discourse. In this study, we focus on both vaccine sentiments equally to understand the nature of public involvement. Our findings provide insights into which *topics* should be addressed to improve vaccine hesitancy and which *actors* can be leveraged as a conduit through which vaccine-related scientific information can be spread. Public health and vaccine communicators can be more proactive in monitoring and understanding these types of users to curb the negative influence of misinformation or misleading claims.

## Supporting information

**S1 Appendix. Keywords.**
(DOCX)

**S2 Appendix. Codebook for positive and negative vaccine discourse on Twitter.**
(DOCX)

**S3 Appendix. The fine-tuned BERT classification accuracy and Area Under Curve (AUC) (S3 Appendix table) and the receiver operating characteristic curve (ROC) on the test set by positive and negative vaccine discourse (S1 Fig).**
(DOCX)

**S4 Appendix. The top 100 most mentioned accounts in positive and negative vaccine twitter discourses.**
(DOCX)

**S1 Fig.**
(TIF)

## Author Contributions

**Conceptualization:** Juwon Hwang, Min-Hsin Su, Xiaoya Jiang.

**Data curation:** Juwon Hwang, Min-Hsin Su, Xiaoya Jiang, Ruixue Lian.

**Formal analysis:** Juwon Hwang, Min-Hsin Su.

**Funding acquisition:** Dhavan Shah.

**Investigation:** Juwon Hwang, Min-Hsin Su.

**Methodology:** Juwon Hwang, Min-Hsin Su, Xiaoya Jiang, Ruixue Lian, Arina Tveleneva.

**Project administration:** Juwon Hwang.

**Software:** Min-Hsin Su, Ruixue Lian.

**Supervision:** Juwon Hwang, Dhavan Shah.

**Validation:** Juwon Hwang, Min-Hsin Su, Xiaoya Jiang, Ruixue Lian.

**Visualization:** Juwon Hwang, Min-Hsin Su, Ruixue Lian.

**Writing – original draft:** Juwon Hwang, Min-Hsin Su, Xiaoya Jiang.

**Writing – review & editing:** Juwon Hwang, Min-Hsin Su, Xiaoya Jiang, Dhavan Shah.

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
