## [Decision Letter · Decision Letter 0]

2 Dec 2021

PONE-D-21-31719Vaccine discourse during the onset of the COVID-19 pandemic:

Thematic structure and source patterns informing efforts to combat vaccine hesitancyPLOS ONE

Dear Dr. Hwang,

Thank you for submitting your manuscript to PLOS ONE. After careful consideration, we feel that it has merit but does not fully meet PLOS ONE’s publication criteria as it currently stands. Therefore, we invite you to submit a revised version of the manuscript that addresses the points raised during the review process.

We look forward to receiving your revised manuscript.

Kind regards,

Kazutoshi Sasahara

Academic Editor

PLOS ONE

Journal Requirements:

" ext-link-type="uri" xlink:type="simple">https://journals.plos.org/plosone/s/file?id=ba62/PLOSOne_formatting_sample_title_authors_affiliations.pdf"

Additional Editor Comments:

Both reviewers think the manuscript is important, but they also think that it needs more improvements. Please read the comments carefully and address them in the revised paper.

Reviewers' comments:

Reviewer's Responses to Questions

**Comments to the Author**

1. Is the manuscript technically sound, and do the data support the conclusions?

Reviewer #1: Yes

Reviewer #2: Partly

2. Has the statistical analysis been performed appropriately and rigorously? 

Reviewer #1: Yes

Reviewer #2: No

3. Have the authors made all data underlying the findings in their manuscript fully available?

Reviewer #1: Yes

Reviewer #2: Yes

4. Is the manuscript presented in an intelligible fashion and written in standard English?

Reviewer #1: Yes

Reviewer #2: Yes

5. Review Comments to the Author

Reviewer #1: This is a strong study that provides important insights into communication flows on Twitter during the early part of the COVID pandemic. It shows key differences between negative and positive sentiment through a comprehensive analysis (i.e., using multiple methods) of original tweets. The methods are sound, and findings support the conclusions. There also is real value here for public health communication.

My main recommendation is to tighten and revise the structure of the article in order to clearly draw out the core insights and to show why these methods are required to generate those insights.

In the "Descriptive" section, it is somewhat difficult for the reader to understand how each part of this sequentially presented analysis builds to the article's findings and main argument. Why, for example, does this section of the article start by tracking the volume of tweets? What does this tell us about COVID communication flows that will be integral to the study's main argument?

Each of these sections in the "Descriptive" section should be introduced with a clear statement of why the authors are choosing to, in order, analyze:

1) Volume of tweets

2) Prevalence of topics

3) Topic communities

4) Relative volume of topics over time

5) Distinct topical prevalence, positive vs. negative valence

6) Same as 5, but in relationship to hashtags

7) Mention network

8) Mention by account type

In other words, this is a richly detailed analysis that needs to be strung together in a more coherent fashion that, in each case along the way, builds and supports a clear argument based on the findings.

More generally, the findings need to be more closely stitched together with the very clear enumeration of issues of 1) content, and 2) actors on pp. 27-28. Revising the article to support, step-by-step, this dual focus on content and actors will help bring more coherence to the methods and findings.

It also would be nice to see a concise, brief summary of the study's three principal takeaways in the first paragraph of the Discussion. This could be one or two sentences following the mention of topics and actors, which then would lead into the more detailed elaborations of each individual finding below.

Also helpful for maximizing the article's impact would be noting present public health communication strategies designed to counter vaccine hesitancy. These could be added to the introduction's literature review, and they would help clarify what is new about the author's conclusions and recommendations for improving these strategies and developing new strategies.

A few other issues:

The authors use the terms "themes," "thematic patterns," and "topics" interchangeably. I suggest sticking to a single term, such as "topic," unless there is a distinction between themes and topics that is important to the study. If so, this distinction needs to be clarified.

The methods arise in mid-paragraph on p. 5. All references to methods to be contained in the methods section.

Something is missing in the first sentence on p. 17. "To understand how the discourses" what?

In Footnote 3 on p. 20, please clarify that @realdonaldtrump is former President Trump's Twitter account. As it now reads, "his" is a pronoun without a first reference.

p. 11: Figure 1 has a typo in the title: "Volumn"

p. 23: "Principle findings" should be "Principal findings"

p. 24 and p. 29: the term "anti-vaxxers" is used here, without definition, though the term implies that all of these social media users were actively opposed to vaccinations. However, we know that there is a range of negative vaccine sentiments, from hesitancy and doubt, to criticism and outright opposition. Accordingly, the literature in this area tends to invoke a spectrum of vaccine confidence (see Larson, 2020; MacDonald the SAGE Working Group on Vaccine Hesitancy, 2015), and the authors open the article by referring to this spectrum. It then is confusing why the term "anti-vaxxers" appears at this point in the article. Please clarify or revise the terminology to align with the article's introduction.

Reviewer #2: Overall

The general area of research is topical. Overall I don't find this paper compelling, partly due to the methods, and partly due to the execution. I am not an expert in STM, so I cannot comment on that. If they could do the paper without machine classification, I think it would be improved, as I don't think these methods are successful on Twitter data--based on their results and my personal experiences.

Major comments

I have done extensive work using machine learning and Twitter, and have never published it because the results are always disappointing. My experience is that 140 characters is not sufficient for proper classification of most Twitter content. The authors used a machine learning method that has, in my view, high error, reporting only 71% and 75% accuracy. The ROC curves in vaccine 3 confirms fairly high combined error tradeoff between sensitivity and specificity. The fact that that 20-30% of the content is coded wrong should cause the authors to be very cautious in the interpretation of their results, and I think presents a serious challenge to publishing this paper. The use of regression models in analyzing an outcome with large measurement error (Table 2) is especially problematic.

I don't know much about structural topic modelling, but these kinds of approaches are more effective when there is more text available. Again, Twitter data presents a challenge here; gathering a meaning from small quantities of text is hard. However, since i am not an expert in this method, I cannot comment further.

Finally, I don't see the practical implications of the paper. The authors claim that "our results highlight the type of content that needs to be addressed to improve vaccine hesitancy" but offer no actual evidence of this. This is a desscriptive paper that does not drill into what imapats vaccine hesitancy. Moreover, changes in attitudes towards covid-19 vaccines render some of these findings somewhat obsolete; what concerns people today (in late 2021) may be different from what concerned them early in vaccine uptake.

Minor comments

Page 4 "social media content does not undergo an editorial processes or scientific vetting". This statement is not true. Twitter, Facebook and YouTube implemented editorial controls specific to Covid and vaccination in 2020.

Page 6. Is the use of Synthesio and web scraping of Twitter legal? Is the use of this tool covered under fair use legislation/practice? Do we know that Synthesesio actually generates true and unbiased samples? Was this service paid for by the authors? Details here are important for the reader for a number of reasons.

6. PLOS authors have the option to publish the peer review history of their article (what does this mean?). If published, this will include your full peer review and any attached files.

Reviewer #1: No

Reviewer #2: No

---

## [Author Response · Author response to Decision Letter 0]

10 Jan 2022

Dear Editor,

We appreciate the feedback from the reviewers and the opportunity to make further revisions on our manuscript for PLOS ONE. We have implemented all of the reviewers’ suggestions and believe that the manuscript is much strengthened as a result. Below, we list the feedback we received from the reviewers, and detail how we have responded to each concern in our revision of the manuscript.

Reviewer #1: This is a strong study that provides important insights into communication flows on Twitter during the early part of the COVID pandemic. It shows key differences between negative and positive sentiment through a comprehensive analysis (i.e., using multiple methods) of original tweets. The methods are sound, and findings support the conclusions. There also is real value here for public health communication.

My main recommendation is to tighten and revise the structure of the article in order to clearly draw out the core insights and to show why these methods are required to generate those insights.

Response: We appreciate the reviewer’s constructive comment. This important comment led us to revise our paper in a way that delivers our main points more effectively. To address this comment, in the introduction, we have now clarified our dual research goals – to identify patterns in the “content” of vaccine discourses on the one hand and to key “actors” with whom users engaging with the vaccine conversation interacted on the other. This revision is presented on page 6:

In the "Descriptive" section, it is somewhat difficult for the reader to understand how each part of this sequentially presented analysis builds to the article's findings and main argument. Why, for example, does this section of the article start by tracking the volume of tweets? What does this tell us about COVID communication flows that will be integral to the study's main argument?

Response: Thank you for this comment. We agree with the reviewer that each finding needs to be presented in a more coherent way and strung together to show the main line of our arguments. Thus, we have now clarified why we started by presenting the volume of tweets and how it would lead readers to the next findings. This revision is presented on page 10.

“We start by plotting the temporal dynamics of twitter volume during the course of our study time, with all types of tweets included, because the overall volume of tweets and how it evolved over time provides a big picture of our dataset before we delve into the examination of content of tweets.”

Each of these sections in the "Descriptive" section should be introduced with a clear statement of why the authors are choosing to, in order, analyze:

1) Volume of tweets

2) Prevalence of topics

3) Topic communities

4) Relative volume of topics over time

5) Distinct topical prevalence, positive vs. negative valence

6) Same as 5, but in relationship to hashtags

7) Mention network

8) Mention by account type

In other words, this is a richly detailed analysis that needs to be strung together in a more coherent fashion that, in each case along the way, builds and supports a clear argument based on the findings.

Response: We appreciate this constructive feedback. The reviewer’s comment truly led us to re-organize our findings in an effective way. We have now made two changes in the results section. First, we have broken down the results section into two major sections (3.1 Content and 3.2 Actors). Second, under each section, we have assigned six subsections for 3.1 Content (3.1.1 Volume of tweet, to 3.1.6 Distinct topical prevalence in relation to hashtags) and two subsections for 3.2 Actors (3.2.1 Mention network and 3.2.2 Mention by account type). This revision allows our findings to be more clearly presented. We thank the reviewer for this very constructive comment. 

More generally, the findings need to be more closely stitched together with the very clear enumeration of issues of 1) content, and 2) actors on pp. 27-28. Revising the article to support, step-by-step, this dual focus on content and actors will help bring more coherence to the methods and findings.

Response: We appreciate this feedback. As stated in the previous responses, we have now clarified our dual purposes for this paper and consistently emphasized them throughout the manuscript in the introduction, methods, results, and discussion. 

It also would be nice to see a concise, brief summary of the study's three principal takeaways in the first paragraph of the Discussion. This could be one or two sentences following the mention of topics and actors, which then would lead into the more detailed elaborations of each individual finding below.

Response: Thank you for this comment. We agree with the reviewer that it would be helpful to briefly summarize our study’s principal takeaways in the first paragraph of the discussion. Thus, we have now included a summary before delving into the detailed discussion of each finding:

“Our study examined a) content difference in positive versus negative tweets surrounding the COVID-19 vaccine, as well as b) prominent actors frequently mentioned by the two camps. The primary findings were summarized as follows: While positive vaccine discussions demonstrated a wider range of perspectives informed by research and medical professionals, negative vaccine conversation was dominated by a narrower set of narratives, such as monetary motivations behind mandatory vaccines, concerns over side effects, as well as conspiratorial beliefs. Furthermore, while positive vaccine discourse interacted with verified agents, negative vaccine discourse tended to mention online influencers and suspended accounts, suggesting possible pathways through which misinformation was spreading.”

Also helpful for maximizing the article's impact would be noting present public health communication strategies designed to counter vaccine hesitancy. These could be added to the introduction's literature review, and they would help clarify what is new about the author's conclusions and recommendations for improving these strategies and developing new strategies.

Response: Thank you for the important comment. We have now created a separate paragraph that notes the present public health communication strategies and shows how our study can provide practical implications based on the current approach. The paragraph at the end of the introduction on page 6 reads:

“By doing so, our study provides important insights and practical guides for public health communicators, which, to date, predominantly focus on disseminating vaccination guidelines or schedules, rather than proactively responding to vaccine hesitancy. Thus, we suggest public health communication professionals should broaden their messages to address specific concerns as revealed in public conversations on social media, as well as leverage a diversity of actors who have been centrally situated in the vaccine-hesitant communities.” 

A few other issues:

The authors use the terms "themes," "thematic patterns," and "topics" interchangeably. I suggest sticking to a single term, such as "topic," unless there is a distinction between themes and topics that is important to the study. If so, this distinction needs to be clarified.

Response: Thank you for the important comment. Indeed, we should stick to a single term, which describes our measurement accurately. We have now used “topics” consistently and removed all the “themes.” We have also revised our title accordingly. We appreciate the reviewer’s comment. 

The methods arise in mid-paragraph on p. 5. All references to methods to be contained in the methods section.

Response: We thank the reviewer for this comment. We have now removed any mention about methods from the introduction. We have specifically removed our statement about the five categorizations of account mentions from this section and moved it to the method section on page 10. We appreciate the reviewers’ comment. 

Something is missing in the first sentence on p. 17. "To understand how the discourses" what?

Response: We have corrected this typo.

In Footnote 3 on p. 20, please clarify that @realdonaldtrump is former President Trump's Twitter account. As it now reads, "his" is a pronoun without a first reference.

Response: Thanks for this comment. We have now clarified that this account is the former President Trump’s in the footnote 3 on page 21.

p. 11: Figure 1 has a typo in the title: "Volumn"

Response: Thanks for the attention to detail. We have corrected this typo.

p. 23: "Principle findings" should be "Principal findings"

Response: We have corrected this typo.

p. 24 and p. 29: the term "anti-vaxxers" is used here, without definition, though the term implies that all of these social media users were actively opposed to vaccinations. However, we know that there is a range of negative vaccine sentiments, from hesitancy and doubt, to criticism and outright opposition. Accordingly, the literature in this area tends to invoke a spectrum of vaccine confidence (see Larson, 2020; MacDonald the SAGE Working Group on Vaccine Hesitancy, 2015), and the authors open the article by referring to this spectrum. It then is confusing why the term "anti-vaxxers" appears at this point in the article. Please clarify or revise the terminology to align with the article's introduction.

Response: We appreciate this important comment. We agree that the appearance of the term, “anti-vaxxers” in this stage of the paper might confuse the readers for several reasons mentioned by the reviewer. We also agree that it is inappropriate to mention this term without definition, since it may lead readers to overlook the dynamic of negative attitudes toward vaccines. We have now removed this term and replaced it with “users with negative attitude toward the COVID-19 vaccine.” We have also made sure that the expression is consistently used throughout the manuscript (e.g., positive/negative attitude toward COVID-19 vaccine) to properly reflect the nuanced nature of this spectrum in negative attitudes, as the reviewer well-pointed out. We thank the reviewer for this comment. 

---

Reviewer #2: Overall

The general area of research is topical. Overall I don't find this paper compelling, partly due to the methods, and partly due to the execution. I am not an expert in STM, so I cannot comment on that. If they could do the paper without machine classification, I think it would be improved, as I don't think these methods are successful on Twitter data--based on their results and my personal experiences.

Major comments

I have done extensive work using machine learning and Twitter, and have never published it because the results are always disappointing. My experience is that 140 characters is not sufficient for proper classification of most Twitter content. The authors used a machine learning method that has, in my view, high error, reporting only 71% and 75% accuracy. The ROC curves in vaccine 3 confirms fairly high combined error tradeoff between sensitivity and specificity. The fact that that 20-30% of the content is coded wrong should cause the authors to be very cautious in the interpretation of their results, and I think presents a serious challenge to publishing this paper. The use of regression models in analyzing an outcome with large measurement error (Table 2) is especially problematic.

Response: Thanks for the reviewer’s important comment. We resonate with the reviewer’s concerns that conducting machine learning, and especially with Twitter data, is not easy. As with other methodologies, machine learning-based approaches certainly have their limitations, as we acknowledged in the manuscript. Nevertheless, machine-learning approach provides a scalable solution to text classification tasks with consistent criteria, adding important insights to our existing knowledge on the topic. There are several advantages of our current approach. First, compared to other unsupervised textual analysis techniques, machine-learning allows tracking discourses that are theoretically important or comparing discourses between groups on the same criteria (e.g., discourses about anti-COVID-19 vaccine). Similarly, when compared with other research methods such as survey, machine-learning approach helps to detect naturally occurring public expression of certain topic or sentiment unobtrusively. 

We also want to emphasize that there are a significant number of published papers that used machine-learning method with Twitter data (Below we listed a few examples). 

Please also let us note that the classification solutions were validated with several additional procedures, including manual coding and structural topic modeling. The accuracy level (71% and 75%) also seems acceptable in our downstream application task, as indicated in other published work (e.g., 74.1% in Zhu et al.,2020).

Lastly, as for the regression models in Table 2, please let us clarify that the numbers are coefficients in the regression model, which is not an index of measurement error. 

Allen, C., Tsou, M. H., Aslam, A., Nagel, A., Gawron, J. M. (2016). Applying GIS and machine learning methods to Twitter data for multiscale surveillance of influenza. PloS one, 11(7), e0157734.

SteelFisher, G. K., Blendon, R. J., Caporello, H. (2021). An uncertain public—encouraging acceptance of Covid-19 vaccines. New England Journal of Medicine, 384(16), 1483-1487.

Xue, H., Bai, Y., Hu, H., Liang, H. (2019). Regional level influenza study based on Twitter and machine learning method. PloS one, 14(4), e0215600.

Zhu, J. M., Sarker, A., Gollust, S., Merchant, R., Grande, D. (2020). Characteristics of Twitter use by state medicaid programs in the United States: machine learning approach. Journal of medical Internet research, 22(8), e18401.

I don't know much about structural topic modelling, but these kinds of approaches are more effective when there is more text available. Again, Twitter data presents a challenge here; gathering a meaning from small quantities of text is hard. However, since i am not an expert in this method, I cannot comment further.

Response: Thank you for this comment. We observe the research communities that are still in the process of advancing algorithms for detecting topics in Twitter data. Nevertheless, structural topic modeling and topic modeling, while not ideal as with any other method, is an established approach for analyzing Tweets. Below are some relevant publications that use structural topic modeling for twitter data analysis.

Surian, D., Nguyen, D. Q., Kennedy, G., Johnson, M., Coiera, E., Dunn, A. G. (2016). Characterizing Twitter discussions about HPV vaccines using topic modeling and community detection. Journal of medical Internet research, 18(8), e6045.

Mishler, A., Crabb, E. S., Paletz, S., Hefright, B., Golonka, E. (2015, August). Using structural topic modeling to detect events and cluster Twitter users in the Ukrainian crisis. In International Conference on Human-Computer Interaction (pp. 639-644). Springer, Cham.

Finally, I don't see the practical implications of the paper. The authors claim that "our results highlight the type of content that needs to be addressed to improve vaccine hesitancy" but offer no actual evidence of this. This is a desscriptive paper that does not drill into what imapats vaccine hesitancy. Moreover, changes in attitudes towards covid-19 vaccines render some of these findings somewhat obsolete; what concerns people today (in late 2021) may be different from what concerned them early in vaccine uptake.

Response: Thank you for the important comment. As the reviewer pointed out, this paper is not providing direct evidence that certain content can reduce vaccine hesitancy. However, our findings do identify specific areas of concerns that can serve as leverage points to create more effective and tailored messages. We believe this is where our practical implications are. 

We also agree with reviewer’s comment that our results from the early phase of the pandemic might be different from the results from the current phase. Our goal of this paper, however, is to highlight vaccine discourses particularly occurring in the early stage of the pandemic. It is reasonable to expect that, when the pandemic began or when the COVID-19 vaccine first became available, the fear or levels of uncertainty about COVID-19 vaccine was unprecedented. It was also the time where all kinds of mis- and disinformation have begun to gain steam. Thus, we intended to detect topical themes, topic prevalence, and network communities regarding pro-vaccine and anti-vaccine discourses in this particular stage. On the other hand, it was also a critical point where health professionals can intervene to shape public perception and spread science-based messages. As such, it is our belief that exploring the topics and prominent actors in this critical period has values in and outside the Covid-19 context. Per the reviewers’ comment, we have now clarified our practical implication in the introduction on page 6:

“By doing so, our study provides important insights and practical guides for public health communicators, which, to date, predominantly focus on disseminating vaccination guidelines or schedules, rather than proactively responding to vaccine hesitancy. Thus, we suggest public health communication professionals should broaden their messages to address specific concerns as revealed in public conversations on social media, as well as leverage a diversity of actors who have been centrally situated in the vaccine-hesitant communities.”

Minor comments

Page 4 "social media content does not undergo an editorial processes or scientific vetting". This statement is not true. Twitter, Facebook and YouTube implemented editorial controls specific to Covid and vaccination in 2020.

Response: Thank you for the comment. Here, we wanted to emphasize that the social media contents, as expressions from individual users, do not generally go through editorial processes before entering the public sphere. Different from other types of contents such as news articles, social media contents represent individuals’ opinions in a more direct way, and with fewer constraints. As the reviewer pointed out, some platforms implemented editorial controls such as rules to curb the spread of misinformation. Nevertheless, social media contents still reflect individuals’ expression primarily because the contents are only subject to the editorial control (i.e., removal) after, not prior to production. More importantly, a variety of opinions still have presence after the editorial control. For example, though Twitter has policies in place to remove tweets containing false claims about COVID-19, there is a fine line to draw between "undisputed" scientific facts and freedom of expression. As a result, strong sentiment or opinions that contradict official recommendations or against the best available scientific evidence were often left to thrive online. Due to the fundamental difference in epistemic production, social media contents like tweets are not bound by common newsroom practice, leading to an "unedited" public sphere (Bimber Gil de Zúñiga, 2020, p.700). 

Bimber, B., Gil de Zúñiga, H. (2020). The unedited public sphere. New Media Society, 22(4), 700-715.

We also referred to Twitter’s COVID-19 misleading information policy (https://help.twitter.com/en/rules-and-policies/medical-misinformation-policy).

Thus, per the reviewer’s comment, we have now clarified our sentences:

“In contrast to traditional news media or scholarly outlets, social media content does not undergo an editorial processes or scientific vetting unless violate certain rules of a platform, thereby allowing users to voice their opinions on their own terms in most cases (p.4).

Page 6. Is the use of Synthesio and web scraping of Twitter legal? Is the use of this tool covered under fair use legislation/practice? Do we know that Synthesesio actually generates true and unbiased samples? Was this service paid for by the authors? Details here are important for the reader for a number of reasons.

Response: Thank you for the comment. Synthesio is a widely used, paid platform to obtain social media data. It is a legal third-party platform and we were authorized to use the platform by paying for it. It is a random sample. While it is possible that data quality is compromised by restrictions such as caps on the maximum return or the total daily queries allowed to run, our data is not obviously biased since Synthesio uses random sampling. Please refer to this website for more information about Synthesio’s dashboard for social media: https://www.synthesio.com/products/social-listening/. 

We have clarified that it is a random sample on page 6: 

“We first scraped Twitter data through Synthesio, a social media monitoring tool to randomly sample 1% of public tweets using COVID-19 related keywords.”

We appreciate all the constructive comments, and we believe we have strengthened our paper considerably by addressing them.

---

## [Decision Letter · Decision Letter 1]

7 Mar 2022

PONE-D-21-31719R1Vaccine discourse during the onset of the COVID-19 pandemic:

Topical structure and source patterns informing efforts to combat vaccine hesitancyPLOS ONE

Dear Dr. Hwang,

Thank you for submitting your manuscript to PLOS ONE. After careful consideration, we feel that it has merit but does not fully meet PLOS ONE’s publication criteria as it currently stands. Therefore, we invite you to submit a revised version of the manuscript that addresses the points raised during the review process.

If applicable, we recommend that you deposit your laboratory protocols in protocols.io to enhance the reproducibility of your results. Protocols.io assigns your protocol its own identifier (DOI) so that it can be cited independently in the future. For instructions see: https://journals.plos.org/plosone/s/submission-guidelines#loc-laboratory-protocols. Additionally, PLOS ONE offers an option for publishing peer-reviewed Lab Protocol articles, which describe protocols hosted on protocols.io. Read more information on sharing protocols at https://plos.org/protocols?utm_medium=editorial-emailutm_source=authorlettersutm_campaign=protocols.

We look forward to receiving your revised manuscript.

Kind regards,

Kazutoshi Sasahara

Academic Editor

PLOS ONE

Journal Requirements:

Additional Editor Comments:

Both reviewers think that the manuscript has been improved, but they also think that it needs minor revisions. Please revised based on the comments.

Reviewers' comments:

Reviewer's Responses to Questions

**Comments to the Author**

1. If the authors have adequately addressed your comments raised in a previous round of review and you feel that this manuscript is now acceptable for publication, you may indicate that here to bypass the “Comments to the Author” section, enter your conflict of interest statement in the “Confidential to Editor” section, and submit your "Accept" recommendation.

Reviewer #1: All comments have been addressed

Reviewer #3: All comments have been addressed

2. Is the manuscript technically sound, and do the data support the conclusions?

Reviewer #1: Yes

Reviewer #3: Yes

3. Has the statistical analysis been performed appropriately and rigorously? 

Reviewer #1: Yes

Reviewer #3: Yes

4. Have the authors made all data underlying the findings in their manuscript fully available?

Reviewer #1: Yes

Reviewer #3: Yes

5. Is the manuscript presented in an intelligible fashion and written in standard English?

Reviewer #1: Yes

Reviewer #3: Yes

6. Review Comments to the Author

Reviewer #1: The authors have improved the readability of the article by providing effective subheadings, by more strongly emphasizing the two-fold nature of the analysis (content/actors), and by including additional cites and cautions regarding the methods. I do think that these methods are appropriate for Twitter analysis, though they must be presented with caution, as the authors do.

My only remaining question concerns the authors' recommendation of paying for the promotion of content among vaccination hesitant Twitter users, as I am unclear of ethics considerations or other potential concerns surrounding such an intervention. Are there cites that could be included to support this recommendation or at least to point out relevant debates concerning it?

Reviewer #3: Authors explore public discourse about covid 19 via random sample of twitter and analyzed using Structural Topic Modeling and Network Analysis.

I am satisfied by most of responses provided by the authors on the reviewes. However, I would argue that authors still need to highlight/defend their selection of methods. Since this same study can be conducted using multiple methods in unsupervised learning, and there is much research available for this as well, it is important that authors explain their strategy of selecting this method. I,e sentiment could also capture using other NLP methods, authors may have a reason to use the ML model instead.

At the same time, it would be helpful to know how did you ended up 2 coders, and were they experienced researchers or external from this study – this will help up to navigate the validity of the codes.

I am not sure how did authors classify the mention network into several subjects – were these done by authors? 1 author? inductively or deductively?

Topical associate represents in the figure 3 needs bit more details – how the association was derived between topics. May be one example to reflect the association would be more helpful to the reader.

Near the figure 4 where authors present the temporal volume, authors state “…While some topics received more sustained overtime attention (e.g., Topic 6 – Conspiracy Theory), others showed a higher degree of fluctuation in Twitter volume (e.g., Topic 8 – Vaccine Development).” – It is not possible to derive this statement from figure 4, it should be explained using figure 3 and it need to bring the connection to figure 4.

Figure 6 is not words its #tags right, ( words can be misunderstood as words in the topic,but this not the case right? - “Words featuring most prominently in positive versus negative vaccine discourses”..

Besides word cloud seems not much providing the context to what authors trying to explain in this. So its more taking space without much value.

Overall, I also felt the need of the coherent story and the key findings in the discussions and authors somewhat have addressed that, but this is still could be improved.

Overall, I enjoyed the implication part the most, which is the essence of this this contribution, and with that I think this paper has a substantial value to the journal.

7. PLOS authors have the option to publish the peer review history of their article (what does this mean?). If published, this will include your full peer review and any attached files.

Reviewer #1: No

Reviewer #3: No

---

## [Author Response · Author response to Decision Letter 1]

14 May 2022

Dear Editor,

We appreciate the feedback we received from the reviewers and the opportunity to make further revisions on our manuscript for PLOS ONE. We have implemented all of the reviewers’ suggestions and believe that the manuscript is much strengthened as a result. Thank you. Below, we list the reviewers’ feedback, and detail how we have responded to each concern in our revision of the manuscript.

Reviewer #1: The authors have improved the readability of the article by providing effective subheadings, by more strongly emphasizing the two-fold nature of the analysis (content/actors), and by including additional cites and cautions regarding the methods. I do think that these methods are appropriate for Twitter analysis, though they must be presented with caution, as the authors do.

My only remaining question concerns the authors' recommendation of paying for the promotion of content among vaccination hesitant Twitter users, as I am unclear of ethics considerations or other potential concerns surrounding such an intervention. Are there cites that could be included to support this recommendation or at least to point out relevant debates concerning it?

Response: We appreciate the reviewer’s important comment. We originally included this practical suggestion by observing the increasing efforts to reduce misleading information policies, which are very creative and proactive. This recommendation, however, should be suggested with caution because it could produce potential ethical issues, as the reviewer pointed out. Thus, after careful consideration, we have removed this recommendation on page 25. We are very grateful to the reviewer for the attention to detail. 

Reviewer #3: Authors explore public discourse about covid 19 via random sample of twitter and analyzed using Structural Topic Modeling and Network Analysis.

I am satisfied by most of responses provided by the authors on the reviewes. However, I would argue that authors still need to highlight/defend their selection of methods. Since this same study can be conducted using multiple methods in unsupervised learning, and there is much research available for this as well, it is important that authors explain their strategy of selecting this method. I,e sentiment could also capture using other NLP methods, authors may have a reason to use the ML model instead.

Response: Thank you for the constructive comment. We agree with the reviewer that a justification on our selection of supervised machine learning will further strengthen our manuscript. Thus, we have added the following paragraph on pages 7-8:

“Specifically, we used supervised machine-learning method, which allows labeling a tweet as expressing positive or negative sentiment regarding COVID-19 vaccines, and in turn, classifying the rest of the tweets based on the labeled samples using the Bidirectional Encoder Representations from Transformers (BERT) machine learning algorithm. There are several advantages with our current approach. Compared to other unsupervised textual analysis techniques, supervised machine-learning allows tracking discourses that are theoretically important or comparing discourses between groups on the same criteria. In our case, our supervised machine learning approach provides a computationally efficient way to classify tweets based on target-specific sentiments (i.e., sentiment toward vaccines), rather than more generic tasks of sentiment detection. By contrast, unsupervised machine learning is to learn the inherent structure of the data without pre-provided labels [40,41]. Since the performance of the unsupervised learning-based sentiment analysis methods heavily relies on the pre-generated lexicon and corpus, if the sentiment/emotion relevant words do not exist in the lexicon or corpus, the classification accuracy will be low [40,41]. Given that COVID-19 and its vaccines involve high levels of uncertainty and novelty, as well as its nature of constantly evolving, we believe that a supervised machine-learning approach would be appropriate to provide a scalable solution to text classification tasks with pre-defined criteria.” 

[40] Hu, T., Wang, S., Luo, W., Zhang, M., Huang, X., Yan, Y., ... Li, Z. (2021). Revealing Public Opinion Towards COVID-19 Vaccines With Twitter Data in the United States: Spatiotemporal Perspective. Journal of Medical Internet Research, 23(9), e30854.

[41] Wang, S., Huang, X., Hu, T., Zhang, M., Li, Z., Ning, H., ... Li, X. (2022). The times, they are a-changin’: tracking shifts in mental health signals from early phase to later phase of the COVID-19 pandemic in Australia. BMJ Global Health, 7(1), e007081.

At the same time, it would be helpful to know how did you ended up 2 coders, and were they experienced researchers or external from this study – this will help up to navigate the validity of the codes.

Response: Thank you for the comment. We had exerted extra cautions when inviting two coders to increase validity of codes as well as reduce any bias. We had two criteria. First, we intentionally excluded experienced researchers who conceptualized and designed this study from the coder pool to prevent any potential bias in codes. Second, familiarity with the primary concept and basic background or experience in content coding are of importance. Based on these criteria, two graduate assistants familiar with the domain subject and content analysis methodology, but not directly involved in designing this research, served as the coders for this study. We have now clarified this information on page 8.

I am not sure how did authors classify the mention network into several subjects – were these done by authors? 1 author? inductively or deductively?

Response: We appreciate this comment. We created classifications of the mention network both inductively and deductively. On one hand, we created primary categories (e.g., scientist/medical source, media/journalists, political source) based on the literature that emphasizes the degree to which the source provides access to medical expertise (inductive process). This process was designed as iterative where all members in the research team compared notes and validated with actual sample tweets until all major categories important to this study were captured. On the other hand, we identified the top 100 twitter accounts that have been mentioned by other users in our sample (Appendix 4) and manually assigned each a category label based on the literature. In this process, we recognized that there were a substantial number of accounts that didn’t fall into the primary categories; thus, two new categories– online influencer and suspended account– were added (deductive process).” It also came to our attention that one account may be classified into more than one category (e.g., a certified doctor who is also an online influencer); thus, we created a priority to classify accounts (deductive process). All of these procedures were completed based on discussion between authors. The following reference was added:

[39] Hwang, J., Shah, D. V. (2019). Health information sources, perceived vaccination benefits, and maintenance of childhood vaccination schedules. Health Communication, 34(11), 1279-1288.

We have now clarified this information on page 11. We are grateful for the reviewer to provide us with an opportunity to clarify our procedures in the manuscript. 

Topical associate represents in the figure 3 needs bit more details – how the association was derived between topics. May be one example to reflect the association would be more helpful to the reader.

Response: Thank you for the comment. We agree with the reviewer that additional explanation on how to interpret the association using edges is needed. Thus, we added the following paragraph on pages 16-17.

“The association between two topics (i.e., nodes) is reflected by the thickness of the undirected line connecting them (i.e., edge). The edges are weighted, calculated based on cosine similarity, with the weight indicating topic co-occurrence in documents. Specifically, the higher the weight, the more likely that two topics are discussed within a given document. For example, vaccine production is more likely to be talked about alongside other coping strategies, yet it has a low cosine similarity value with the topic of conspiracy [28]." 

Near the figure 4 where authors present the temporal volume, authors state “…While some topics received more sustained overtime attention (e.g., Topic 6 – Conspiracy Theory), others showed a higher degree of fluctuation in Twitter volume (e.g., Topic 8 – Vaccine Development).” – It is not possible to derive this statement from figure 4, it should be explained using figure 3 and it need to bring the connection to figure 4.

Response: Thank you for the comment. We have now added a note in Figure 4 to add clarity of its interpretation as well as to bring connection.

Figure 6 is not words its #tags right, ( words can be misunderstood as words in the topic,but this not the case right? - “Words featuring most prominently in positive versus negative vaccine discourses”.. Besides word cloud seems not much providing the context to what authors trying to explain in this. So its more taking space without much value.

Response: We appreciate your attention to detail. We originally included results about distinct topical prevalence in relation to hashtags to further clarify the difference between positive and negative vaccine discourses. However, we also feel that these results do not provide much novel information; thus, we have decided to remove these results. 

Overall, I also felt the need of the coherent story and the key findings in the discussions and authors somewhat have addressed that, but this is still could be improved.

Overall, I enjoyed the implication part the most, which is the essence of this this contribution, and with that I think this paper has a substantial value to the journal.

We appreciate all the constructive comments. We have gone through the discussion to present our key findings more effectively. We have also tried our best to improve coherency in this section. We believe we have strengthened our paper considerably by addressing the reviewers’ comments. Thank you.

---

## [Decision Letter · Decision Letter 2]

21 Jun 2022

PONE-D-21-31719R2Vaccine discourse during the onset of the COVID-19 pandemic:

Topical structure and source patterns informing efforts to combat vaccine hesitancyPLOS ONE

Dear Dr. Hwang,

Thank you for submitting your manuscript to PLOS ONE. After careful consideration, we feel that it has merit but does not fully meet PLOS ONE’s publication criteria as it currently stands. Therefore, we invite you to submit a revised version of the manuscript that addresses the points raised during the review process. Both reviewers agreed that the manuscript was improved. But one reviewer required a few minor revision. you Please read comments and address accordingly.

We look forward to receiving your revised manuscript.

Kind regards,

Kazutoshi Sasahara

Academic Editor

PLOS ONE

Journal Requirements:

Reviewers' comments:

Reviewer's Responses to Questions

**Comments to the Author**

1. If the authors have adequately addressed your comments raised in a previous round of review and you feel that this manuscript is now acceptable for publication, you may indicate that here to bypass the “Comments to the Author” section, enter your conflict of interest statement in the “Confidential to Editor” section, and submit your "Accept" recommendation.

Reviewer #1: (No Response)

Reviewer #3: All comments have been addressed

2. Is the manuscript technically sound, and do the data support the conclusions?

Reviewer #1: Yes

Reviewer #3: Yes

3. Has the statistical analysis been performed appropriately and rigorously? 

Reviewer #1: Yes

Reviewer #3: Yes

4. Have the authors made all data underlying the findings in their manuscript fully available?

Reviewer #1: Yes

Reviewer #3: (No Response)

5. Is the manuscript presented in an intelligible fashion and written in standard English?

Reviewer #1: Yes

Reviewer #3: Yes

6. Review Comments to the Author

Reviewer #1: This article is much improved and, in my view, needs only a couple minor revisions:

On p. 10: Should "Sandford" be "Stanford"?

p. 25: The authors indicated in their responses to reviewer comments that they had removed references to paying for promotion of content, yet this sentence remains in the article: "Again, paid sponsorship of influencers accounts may prove to be an effective intervention strategy." Do they still intend to include such recommendations?

Reviewer #3: I thank the authos for addressing my comments in the manuscript and also explaining in the response letter. New changes appearing much improved and clear decription of the conduct and results and the story behind the results. I think the insights will bring much value to the readers.

7. PLOS authors have the option to publish the peer review history of their article (what does this mean?). If published, this will include your full peer review and any attached files.

Reviewer #1: No

Reviewer #3: **Yes: **Dilrukshi Gamage

---

## [Author Response · Author response to Decision Letter 2]

27 Jun 2022

Dear Editor,

We appreciate the feedback we received from the reviewers and the opportunity to make further revisions on our manuscript for PLOS ONE. We have implemented all of the reviewers’ suggestions and believe that the manuscript is much strengthened as a result. Thank you. Below, we list the reviewers’ feedback, and detail how we have responded to each concern in our revision of the manuscript.

Reviewer #1: This article is much improved and, in my view, needs only a couple minor revisions:

On p. 10: Should "Sandford" be "Stanford"?

Response: We appreciate your attention to detail. We have corrected this typo and proofread the manuscript again. Once again, we thank the reviewer for the comment.

p. 25: The authors indicated in their responses to reviewer comments that they had removed references to paying for promotion of content, yet this sentence remains in the article: "Again, paid sponsorship of influencers accounts may prove to be an effective intervention strategy." Do they still intend to include such recommendations?

Response: We appreciate the reviewer’s important comment. This recommendation, along with references to paying for promotion of content, should be removed, as it could produce potential ethical issues. We apologize for not getting it right in advance. We have now removed this sentence and are very grateful to the reviewer for the comment. 

Reviewer #3: I thank the authos for addressing my comments in the manuscript and also explaining in the response letter. New changes appearing much improved and clear decription of the conduct and results and the story behind the results. I think the insights will bring much value to the readers.

Response: We appreciate all the constructive comments in the previous round. We believe we have strengthened our paper considerably by addressing the reviewers’ comments. Thank you.

---

## [Editor Report · Decision Letter 3]

30 Jun 2022

Vaccine discourse during the onset of the COVID-19 pandemic:

Topical structure and source patterns informing efforts to combat vaccine hesitancy

PONE-D-21-31719R3

Dear Dr. Hwang,

We’re pleased to inform you that your manuscript has been judged scientifically suitable for publication and will be formally accepted for publication once it meets all outstanding technical requirements.

Kind regards,

Kazutoshi Sasahara

Academic Editor

PLOS ONE

Additional Editor Comments (optional):

Now all the reviewers think it's properly revised.
---

## [Editor Report · Acceptance letter]

4 Jul 2022

PONE-D-21-31719R3 

Vaccine discourse during the onset of the COVID-19 pandemic: Topical structure and source patterns informing efforts to combat vaccine hesitancy 

Dear Dr. Hwang:

I'm pleased to inform you that your manuscript has been deemed suitable for publication in PLOS ONE. Congratulations! Your manuscript is now with our production department. 

Kind regards, 

on behalf of

Dr. Kazutoshi Sasahara 

Academic Editor

PLOS ONE